# Sex Differences in Cardiovascular Diseases: Exploring the Role of Microbiota and Immunity

**DOI:** 10.3390/biomedicines12081645

**Published:** 2024-07-24

**Authors:** Laura Franza, Mario Caldarelli, Emanuele Rocco Villani, Rossella Cianci

**Affiliations:** 1Emergency, Anesthesiological and Reanimation Sciences Department, Fondazione Policlinico Universitario A. Gemelli-IRCCS of Rome, 00168 Rome, Italy; cliodnaghfranza@gmail.com; 2Emergency Department, Azienda Ospedaliero-Universitaria di Modena, Largo del Pozzo, 71, 41125 Modena, Italy; 3Department of Translational Medicine and Surgery, Catholic University of the Sacred Heart, 00168 Rome, Italy; rossella.cianci@unicatt.it; 4Fondazione Policlinico Universitario A. Gemelli, Istituto di Ricovero e Cura a Carattere Scientifico (IRCCS), 00168 Rome, Italy; 5Department of Geriatrics, Orthopedics and Rheumatology, Università Cattolica del Sacro Cuore, 00168 Rome, Italy; emanuelerocco.villani@unicatt.it; 6UOC Geriatra-Disturbi Cognitivi e Demenze, Dipartimento di Cure Primarie, AUSL Modena, 41012 Modena, Italy

**Keywords:** cardiovascular diseases, sex differences, gut microbiota, personalized medicine, inflammation

## Abstract

Cardiovascular diseases (CVDs) are the most common cause of mortality and morbidity in Western countries, thus representing a global health concern. CVDs show different patterns in terms of the prevalence and presentation in men and women. The role of sex hormones has been extensively implicated in these sex-specific differences, due to the presence of the menstrual cycle and menopause in women. Moreover, the gut microbiota (GM) has been implicated in cardiovascular health, considering the growing evidence that it is involved in determining the development of specific diseases. In particular, gut-derived metabolites have been linked to CVDs and kidney disorders, which can in turn promote the progression of CVDs. Considering the differences in the composition of GM between men and women, it is possible that gut microbiota act as a mediator in regard to the sex disparities in CVDs. This narrative review aims to comprehensively review the interplay between sex, GM, and CVDs, discussing potential mechanisms and therapeutic options.

## 1. Introduction

Cardiovascular diseases (CVDs) are the main driver of mortality worldwide, accounting for a substantial proportion of global morbidity and mortality rates, particularly in Western countries. This group of disorders includes a large number of both acute and chronic conditions, which have a varying impact on long-term health [1]. While traditionally perceived as predominantly affecting men, CVDs are increasingly recognized as a significant health threat to women as well [2]. Up until a few decades ago, women were largely excluded from clinical trials, thus there was very little information on the presentation, prevalence, and outcomes of these diseases in this part of the population [3]. However, recently physicians have observed differences in CVDs between men and women; for this reason, markers of cardiovascular diseases, which are largely used in clinical practice both for diagnostic and prognostic purposes, may need different cut-offs for men and women [4]. Another interesting aspect that needs to be considered is that while in men after puberty, hormone fluctuations are not particularly significant, women experience significant variations during every menstrual cycle [5]. Pregnancy and the menopause are also associated with further variations in hormonal levels and specific changes in cardiovascular health [6,7], and women who experience conditions such as polycystic ovary syndrome (PCOS) [8] and endometriosis [9] present a set of risks of their own. Sex differences in CVDs are also present in terms of presentation, which in turn take a toll on mortality and morbidity, as they are associated with delayed diagnosis and treatment; it has been observed that women presenting with CVDs receive worse care and experience worse outcomes overall, when compared to men [10,11].

While the importance of hormonal fluctuations has been identified as one of the key components in determining sex differences in CVDs, recent advances in research concerning the microbiota have highlighted the role of gut microbiota (GM) in cardiovascular health, prompting an investigation into its potential contribution to sex-specific patterns in CVDs [12,13]. Sex-specific differences in microbiota composition have been observed in different disorders [14], and it appears that there may be crosstalk with different hormonal pathways in determining them [15].

This narrative review aims to explore the complex interplay between sex, GM, and CVDs, shedding light on the underlying mechanisms and therapeutic implications.

We used a systematic approach to gather and analyze relevant literature on the topic of cardiovascular diseases, sex differences, and the role of microbiota. A comprehensive search of electronic databases, including PubMed, MEDLINE, Embase, and Google Scholar, was conducted. We searched for keywords, such as “cardiovascular diseases”, “sex differences”, “microbiota”, “gut microbiome”, “inflammation”, “metabolites”, and “personalized medicine”. We only took into consideration articles written in the last ten years that were in English. We considered primary research articles, review articles, meta-analyses, and systematic reviews, and prioritized studies focusing on GM, its composition, function, and its relationship with cardiovascular health. The quality of the included studies was assessed based on the study design, methodology, sample size, and relevance to the research question. Studies with robust methodologies and findings were given more space in the analysis and interpretation of the results. We extracted data from the selected studies and summarized them to comprehensively understand the current knowledge. We also considered key findings related to sex differences in cardiovascular diseases, mechanisms of gut microbiota–sex interactions, and the impact of microbiota-derived metabolites on cardiovascular health. All the extracted data were analyzed to identify common themes, trends, and gaps in the literature. Emphasis was placed on elucidating the mechanisms underlying sex-specific differences in GM composition and function, as well as their implications for cardiovascular diseases.

## 2. Role of Sex Differences in Cardiovascular Risk

Sex disparities in healthcare are a well-known issue and traditional approaches do nothing but promote differences in care [16]. The issue is complex and embedded with other matters, particularly the socio-economic and educational status of the patient [17].

CVDs present variations observed in disease prevalence, risk factor profiles, and clinical outcomes between men and women [18]. Historically, they were thought to primarily affect men, which led to the underrepresentation of women in cardiovascular research and clinical trials [19]. However, epidemiological data has revealed that CVDs are the main cause of death among women globally, highlighting the need for sex-sensitive approaches to cardiovascular care. CVDs in women often present different characteristics to those in men and are the subject of specific guidelines from the American College of Cardiology, published in 2020. One of the key elements highlighted in these guidelines is the role of inflammation in promoting the development of CVDs in women. It is interesting to observe that in this part of the population, there are some sex-specific causes at play, such as adverse pregnancy outcomes, and some causes that are more prevalent in women (e.g., chest wall radiation due to breast cancer, adverse socio-economic conditions, autoimmune disorders) [20,21]. More traditional risk factors for CVD present differently in men and women. Hypertension, for instance, is more commonly observed later in life in women, compared to men, but increases more steeply and does not vary as much between day and night, leading to a higher incidence of cardiovascular events during the night [22]. Women can also experience hypertension during pregnancy or as a consequence of using contraceptive medication [23]. While hypertension usually affects men earlier, the contrary is true for type 2 diabetes (DM2): DM2 is more prevalent in girls than in boys, then it reaches a similar prevalence in later years, which means that women experience DM2 for longer periods, thus increasing the burden of the disease on the endothelium [24]. Overall, as discussed by Appelman et al., CVD risk factors differ significantly between men and women, in general it appears that up until the menopause, women present a more favorable risk profile, which tends to become more similar to men’s as they approach the menopause [25].

In a recent paper by Rosano et al. the specific differences between men and women in the context of heart failure were explored and it was observed that some of them were driven by underlying bias (different availability of therapies, different likelihood of being treated more or less aggressively), others seemed to be determined by intrinsic differences, for instance, in therapy responsiveness, or in regard to disease characteristics [26]. Similar objective differences have also been observed in ischemic heart disease: also, in this case, women present different patterns of presentation, different forms of the disease, and even different risk factors [27]. While in the previous cases, data on women are lacking, due to an underrepresentation in clinical trials, Takotsubo disease presents quite the opposite problem: as highlighted by Nishimura et al., Takotsubo is considered to be a more “feminine” disorder, which appears to be more likely to be triggered by intense physical activity, rather than emotional stress, and studies do not often focus on this group [28].

Moreover, Peters et al., in a study conducted on 171,897 women and 167,993 men, reported that the diagnosis of coronary heart disease is growing in women, who also exhibit higher mortality rates in the short-to-medium term following an MI compared to men. Furthermore, in a cohort of 16,763 premenopausal women, a 10-year follow-up revealed 1569 major adverse cardiac events. By the end of the follow-up, a significant number of these women were overweight, pre-diabetic, or had diabetes mellitus, all conditions associated with over-inflammation [29].

Interesting data have also emerged from the BRECARD study [30]. This research aimed to investigate whether the density of adipose tissue in the breast gland was linked to different rates of major adverse cardiac events (MACEs) in premenopausal women, at the 10-year follow-up. The results showed that women with a higher amount of adipose tissue in the breast gland had a higher rate of MACEs. Therefore, a screening mammography could be recommended for overweight women to assess breast density and predict the risk of MACEs [30]. A possible explanation could be that the breast fat deposit overexpresses inflammatory pathways, leading to worse cardiovascular outcomes. In another study, Sartu et al. evaluated the expression of sodium–glucose transporter 2 (SGLT2), inflammatory cytokines, and sirtuins in breast fat tissue at the baseline, as well as serum cytokines in fatty breasts versus non-fatty breasts in premenopausal women, both at the baseline and after 12 months of follow-up. The study aimed to correlate the expression of SGLT2, cytokines, and sirtuins with other clinical variables, such as intima–media wall thickness (IMT), left ventricle mass (LVM), left ventricle ejection fraction (LVEF), and the myocardial performance index (MPI) [31]. The results showed that women with fatty breasts, compared to those with non-fatty breasts, overexpressed SGLT2 and inflammatory cytokines, while down-regulating breast sirtuins. The expression of SGLT2 and inflammatory cytokines, along with the inverse expression of tissue sirtuin 3 (SIRT-3) and the breast percentage density, were linked to changes in the MPI after one year of follow-up. Fatty breast tissue and SGLT2 were inversely predictive of the normalization of clinical parameters (NCPs), whereas higher levels of SIRT-3 increased the probability of the NCPs at the one-year follow-up [31].

One aspect that also needs to be given sufficient attention is that, overall, the burden of CVDs in women increases after the menopause. This phase is associated with significant changes in the hormonal status of women, in particular a decline in estradiol and progesterone occurs, and an increase in follicle-stimulating hormones has been observed, even though the timeline of these changes can vary greatly [32]. Estradiol is linked to reduced inflammation and the stiffness of endothelial and myocardial tissue, while progesterone promotes vasorelaxation [33]. These hormonal changes are associated with a variety of symptoms, which can be CVD risk factors (e.g., sleep disturbances, mood disorders), and determine cardiometabolic changes. It has been observed that during the menopause, the risk of developing metabolic syndrome increases beyond what would be expected simply because of chronological aging. Also, indexes of vascular health and body composition worsen during this period [34], and it is worth noting that the GM composition also changes [35]. Progesterone, for instance, exerts vasodilatory effects by enhancing nitric oxide production and promoting the relaxation of blood vessels. Additionally, progesterone exhibits anti-inflammatory effects and can influence the renin–angiotensin–aldosterone system, which is crucial for blood pressure regulation. The decrease in progesterone levels during the menopause may worsen cardiovascular changes linked to estrogen deficiency, potentially increasing cardiovascular risk in post-menopausal women [36]. Low testosterone levels are linked to obesity, metabolic syndrome, type 2 diabetes mellitus, and alterations in lipid profiles [37]. Lincoff et al. conducted a multicenter, randomized, double-blind, placebo-controlled trial, recruiting 5246 men aged from 45 to 80 years old, with preexisting cardiovascular disease, or at high risk of cardiovascular disease. Those reporting symptoms of hypogonadism and having two fasting testosterone levels below 300 ng per deciliter were enrolled. Participants were randomly assigned to receive daily transdermal 1.62% testosterone gel (with dose adjustments to maintain the testosterone levels between 350 and 750 ng per deciliter) or the placebo gel. They concluded that in middle-aged and older men with hypogonadism and preexisting cardiovascular disease, or an increased cardiovascular risk, daily treatment with transdermal testosterone for approximately 2 years was found to be noninferior to the placebo regarding the incidence of major adverse cardiac events [38].

The same sex hormones appear to function differently in the cardiovascular cells of men and women. Kararigas et al., in an ex vivo study of myocardial samples from patients with aortic valve stenosis obtained during surgery, showed that several genes in cardiomyocytes are regulated differently in men and women following treatment with estradiol [39]. For instance, estradiol treatment led to the up-regulation of the Myosin regulatory light chain interacting protein (MYLIP) in heart samples and cardiomyocytes from male individuals, but this effect was not observed in samples from female individuals [39]. Estradiol-induced, sex-specific collagen regulation was also observed in human cardiac fibroblasts, suggesting that this regulation is conserved across species [40]. An interesting model that can help understand the role of genetics in cardiovascular health and disease is offered by genetic cardiomyopathies [41]. In the case of hypertrophic cardiomyopathy, for instance, even though the disease is autosomal dominant, men are not as likely as women to present with pathogenic sarcomere variants; also, women of pre-menopausal age appear to be at least partially protected in the presence of myosin-binding protein C3 (MYBPC3) mutations [42]. Yet, there is also evidence that women experience a higher mortality rate overall, confirming that there is an interaction between specific mutations and gender, and something similar can be observed in the case of dilated cardiomyopathy [43].

Another interesting aspect is that the Y chromosome may be linked to specific cardiovascular phenotypes. At the same time, genes encoded on the X chromosome have also been associated with specific cardiovascular phenotypes, yet research still needs to be carried out to understand the exact implications on human health, as the available studies focus on animal models [44,45]. Ronen et al. discovered that among the 227 genes showing differential expression between male and female pluripotent cells, 85 genes contained Sry binding sites, indicating that the Y-specific gene acts as a genetic modulator [46]. Sry, located on the Y chromosome, determines the development of testes rather than ovaries and is the primary cause of the male/female gonadal hormone imbalance, which significantly contributes to general sexual dimorphism [47]. A positive correlation has been reported between men diagnosed with Y polysomy and an increased risk of death from circulatory system diseases [48]. Studies have shown that the risk of coronary artery disease (CAD) is higher in carriers of haplogroup I1 compared to other Y chromosome haplogroups, indicating pleiotropic effects of the Y chromosome on the susceptibility to CAD [49]. The association of the male-specific region of Y chromosome (MSY) genes with risk factors for CVD, including hypertension, circulating total cholesterol, LDL levels, and a paternal history of cardiac diseases, has been demonstrated through single nucleotide polymorphisms [50]. Transcriptome analyses of heart tissues from new-onset heart failure revealed differences in the expression levels of sex chromosome genes. Specifically, Y chromosome-related transcripts, such as USP9Y, DDX3Y, RPS4Y1, and EIF1AY, were found to be overexpressed in males [51].

Interesting new research is taking place on the gradual mosaic loss of the Y chromosome (mLOY). More specifically, mLOY is a common occurrence in elderly men, which it is associated with shorter life expectancy, as well as increased risk of cancer and other disorders [52]. In the context of cardiovascular diseases, studies in both animal models and humans have indicated that mLOY is associated with higher mortality from heart failure. Even among individuals undergoing transcatheter aortic valve replacement, an elevated cardiovascular risk linked to mLOY has been observed [53]. A proposed pathophysiological mechanism linking mLOY to cardiovascular disease is the promotion of fibrosis. This phenomenon could stem from cardiac macrophages derived from Y chromosome-deficient hematopoietic stem cells, which exhibit altered functions. Cardiac macrophages, carrying the loss of the Y chromosome, may infiltrate the heart in response to various cardiac injuries or replace resident yolk sac-derived macrophages, as individuals age. Macrophages with mLOY demonstrate heightened activation of a signaling network that promotes fibrosis [54].

The X chromosome can also influence the phenotypic expression of inflammatory risk factors. These genes encompass those involved in apoptosis, lipid oxidation, and the production of oxygen-derived free radicals by mitochondria [45]. Stamova et al., in an analysis of the RNA levels of 683 genes located on the X chromosome of individuals at specified time intervals following a stroke, observed that X-linked genes exhibited greater and differential up-regulation in women compared to men. In women, the up-regulated genes were associated with the post-translational modification of proteins, small molecule biochemistry, and cell–cell signaling [55].

It is also worth noting that the X chromosome contains 118 microRNAs (miRNAs), which have been confirmed to be involved in cardiac remodeling [56]. Also, while, in general, in those who have two X chromosomes, one of them is silenced, about 15–25% of the genes manage to escape inactivation [57], and it has been observed in murine models that mice with two X chromosomes experience worse cardiac remodeling [58].

## 3. Mechanisms of Sex-Specific Differences in Gut Microbiota Composition

As mentioned above, GM exhibits sex-specific variations, both in terms of its composition and its capabilities, influenced by factors, such as sex hormones, genetic predispositions, and dietary habits.

Intuitively, the association between sex hormones and GM is clear, as suggested by the variations in gastrointestinal (GI) function associated with the menstrual cycle, pregnancy, and the menopause. This association has been studied in murine models and has shown that there are differences in GM composition that appear to be influenced by hormones [59]. It is interesting to observe that GM can also influence sex hormones. For instance, GM is responsible for the transformation of estrogen into its active compounds, through β-glucuronidase. The process can be impaired in the case of dysbiosis, which is in turn promoted by low estrogen levels [60]. Specific taxa are associated with higher estrogen levels, in particular Clostridia and some Ruminococcaceae, yet the association is only present after the menopause [35,61]. Estradiol is associated with immune system activation in the gut, promoting B cell proliferation and interleukin (IL)-12 and interferon (IFN)-γ production, which in turn translates into increased gut permeability, promoting a “leaky” gut condition [62]. Androgens also influence the composition of the GM: similar to estrogens, it increased the bacterial diversity, but did not increase inflammation or gut permeability alterations. If there was an excess of androgens, it would be possible to observe a reduction in variability and a reduction in specific groups of bacteria (e.g., *Akkermansia* and Ruminococcaceae), while some other species increased (e.g., Bacteroides, *Escherichia*/*Shigella*, and Streptococcus). It is worth noting that androgen excess is present, among other conditions, in persons with polycystic ovary syndrome (PCOS) [63].

Dietary patterns and lifestyle factors influence microbial diversity and community structure, further modulating sex-specific microbiota profiles. It has been observed, for instance, that the recommended daily intake of fruit and vegetables is more commonly met by women, which positively influences GM composition [61]. At the same time, even when dietary patterns are similar, the influence on GM composition varies based on sex: a high-fat diet promotes the presence of Lactobacillus, Alistipes, Lachnospiraceae, and Clostridium in males (both in murine and human models), while the same was not observed in females [64]. It seems relevant to underline that women are more likely to engage in diet culture, which may involve fad diets, the consumption of so-called “superfoods”, and elimination of entire food groups, which of itself is likely to negatively impact GM composition [65]. Women are also more likely to exercise [66], but, from a GM composition standpoint, men seem to enjoy more advantages. It is understood that regular physical activity positively influences GM composition [67], but when analyzing the differences between males and females in a murine model, it appeared that GM composition varied significantly only in males [68].

Weight and body mass index (BMI) also share a bidirectional relationship with GM composition: in people who are overweight or obese, Firmicutes, the genus Clostridium, and the species *Eubacterium rectale*, *Clostridium coccoides*, *Lactobacillus reuteri*, *Akkermansia muciniphila*, *Clostridium histolyticum*, and *Staphylococcus aureus*, are more common [69]. However, in a 2016 study, it was observed that the differences in GM composition between those who were at a healthy weight and those who were overweight or obese were also influenced by sex and reflected on other aspects of metabolic health, such as plasma lipids [70]. It is worth noting that the women involved in this study were post-menopausal, to avoid hormonal fluctuations associated with the menstrual cycle. In a study by Kaliannan et al., the authors observed that estrogen appears to be a key mediator of GM composition and a promoter of metabolic syndrome, through inflammation [71]. Yet, the study was carried out on mice and further data on humans are necessary.

Other elements can also influence gut microbiota composition, such as alcohol consumption, stress, and medications, but it is worth noting that all these elements are often socially coded and, thus, vary significantly among men and women. Alcohol consumption, for instance, is perceived as more common and appropriate among men and is responsible for dysbiosis and increased gut permeability [72]. Stress is also capable of influencing the composition of GM: chronic stress can negatively influence the presence of Firmicutes and Tenericutes phyla, in particular Lactobacillaceae and Coprococcus, for instance, while other forms of stress appear to induce different forms of dysbiosis [73]. As discussed above, women are more likely to experience different forms of stress, compared to their male counterparts [74].

Overall, it appears that GM composition also depends on sex, both because of physiological factors and societal perceptions and norms (Figure 1).

## 4. Influence of Sex-Specific Environmental Factors on Gut Microbiota

### 4.1. Smoking and Alcohol Consumption

Both smoking and alcohol consumption are behaviors that are more prevalent in men than in women [75,76]. These variations stem from complex interplays between social, cultural, and biological factors. Social norms often dictate different roles and expectations for men and women regarding substance use. Traditional gender roles may encourage men, for instance, to be more willing to engage in risk-taking actions and express assertiveness, which can lead to higher rates of alcohol consumption and smoking [77]. On the other hand, women who engage in substance use are more likely to face social stigma and judgment, which can lead to a lower prevalence of these behaviors, but can also encourage more dangerous forms of behavior, particularly leading to underreporting or concealment [78]. It is also worth noting that women encounter unique social pressures related to body image and appearance, which can influence their attitudes towards smoking as a means of weight control, while alcohol is associated with weight gain, which can act as a deterrent [79,80]. Gender inequalities in these domains may exacerbate disparities in substance use and related health outcomes.

Moreover, biological variances, including differences in metabolism and hormone levels, can influence how alcohol and tobacco are processed and tolerated by the body [81]. Chronic alcohol consumption has been linked to dysbiosis in GM, characterized by reduced microbial diversity and alterations in specific taxa, such as an increased number of pathogenic bacteria, while beneficial species decrease [82]. In a murine model, it was observed, for instance, that alcohol exposure in males determines a more significant increase in gut permeability, which can be reversed through fecal microbiota transplantation (FMT) from healthy female donors [83]. Moreover, it was observed that FMT presents benefits in regard to alcohol abuse [84].

Comparably, smoking has been associated with changes in gut microbiota [85]. A reduction in Ruminococcaceae was observed in men, which was not observed in women [86].

Nevertheless, the mechanisms that influence the microbiota in both sexes have not yet been clarified; environmental factors, such as smoking and alcohol consumption, will certainly need to be investigated in depth [87].

### 4.2. Diet and Exercise

Diet and physical exercise strongly influence the GM, modifying the qualitative and quantitative composition, intervening in metabolic processes and the immune system. Recent research highlights specific sex differences.

However, studies suggest that the impact of diet and exercise on GM may vary between men and women. For instance, dietary patterns rich in fiber and plant-based foods have been associated with increased microbial diversity and a more favorable GM profile in both sexes [88]. Yet, hormonal fluctuations and metabolic differences between sexes may influence how the gut microbiota responds to specific dietary components and exercise regimens. Additionally, exercise has been shown to exert sex-specific effects on GM composition, potentially due to variations in body composition, hormone levels, and metabolic responses to physical activity [89]. Understanding these sex-specific interactions among diet, exercise, and GM is crucial for developing personalized lifestyle interventions aimed at optimizing microbial balance and promoting overall health and well-being.

### 4.3. Medications

Prescription patterns in men and women can be quite different. In an Italian cross-sectional study from 2017, it was observed that some classes of medication were more likely to be prescribed to men than to women, and vice versa. In the context of cardiovascular disorders, vasodilators, ACE inhibitors, and antiplatelet agents are more commonly prescribed to men [90]. Interestingly, though, women are, overall, more likely to be medicated than men [91]. As discussed, men and women develop cardiovascular disorders at different ages and in association with different patterns [92], yet these elements are unlikely to justify the differences observed. Another aspect that needs to be taken into consideration is that the same medications, used at the same dose, by both sexes have different actions, due to differences in metabolism and pharmacokinetics [93].

The interaction between drugs and microbiota is well-known; on the one hand, the microbiota can modify the way drugs are absorbed and metabolized; on the other hand, medications can alter the microbiota itself [94].

In the case of drugs that act on the cardiovascular system, their use can improve gut health, through adequate vascularization. Yet, patients who suffer from cardiovascular diseases often present dysbiosis, which can prevent the drugs from acting as intended, in a self-perpetuating vicious circle [95].

## 5. Cardiovascular Diseases and Gut Microbiota

The GM consists of a varied population of microorganisms, which includes bacteria, yeast, and viruses [96]. The major phyla of gut microbes include Bacteroidetes, Firmicutes, Actinobacteria, Fusobacteria, Proteobacteria, and Verrucomicrobia, with Firmicutes and Bacteroidetes comprising approximately 90% of the GM [97]. In the interaction between the host and the GM, a fundamental role is played by the intestinal epithelial barrier (IEB), which is made up of a single layer of cells, composed of various epithelial cell types. Below this epithelial layer lies the lamina propria, which is a thin layer of connective tissue, helping to promote effective communication between immune cells and the microbiota [12]. Maintaining a delicate equilibrium between the intestinal barrier, GM, and immune cells is vital for preserving gut homeostasis and overall health within the gastrointestinal tract, which operates as a sophisticated ecosystem. The term “leaky gut” is used to illustrate the dysfunction of the gut barrier [19]. When the intestinal barrier breaks down, it permits bacteria and their chemical products to penetrate the mucosa, initiating uncontrolled inflammatory signaling cascades [12]. The GM plays a crucial role in cardiovascular health, modulating various aspects of CVD pathophysiology, including inflammation, lipid metabolism, and vascular function. There are three main mechanisms through which the microbiota seems to be capable of promoting CVDs, which also influence one another: inflammation, immune dysfunction, and increased barrier permeability [96]. It is worth noting that these phenomena are linked to the presence of dysbiosis.

Dysbiosis is an alteration of the microbial composition and function and is involved in the development and progression of CVDs [98]. In these conditions, GM can indeed promote systemic inflammation, insulin resistance, and dyslipidemia, which are all contributing factors to atherosclerosis and cardiovascular events. In a study by Warmbrunn et al., for instance, it was observed that a lack of Christensenellaceae, Methanobrevibacter, and various Ruminococcaceae was associated with higher lipidaemia and increased CVD risk [99]. Moreover, microbial-derived metabolites, such as trimethylamine N-oxide (TMAO) and short-chain fatty acids (SCFAs), exert profound effects on cardiovascular physiology, further highlighting the intricate interplay between GM and CVDs [100]. At the same time, GM is negatively impacted by poor cardiovascular health. Conditions negatively impacting the cardiovascular system have been linked to poor gut health and GM dysbiosis: poor gut perfusion is associated with the development of a leaky gut, which allows the translocation of microbial byproducts in the bloodstream and impairs the diversity of the GM, promoting dysbiosis [101]. GM alterations and increased gut permeability then promote systemic inflammation, which is in itself a risk factor for CVDs. The relationship between cardiovascular disease and dysbiosis has been observed in different conditions, ranging from atherosclerosis [102], to hypertension [103], myocardial infarction [104], and even arrhythmias [105]. Heart failure (HF) is a prime example of this interaction. Traditionally, it has been defined as an impaired forward flow, often exemplified by low cardiac output [106].

Hemodynamic changes in HF can have significant effects on the intestinal mucosa, resulting in greater intestinal permeability [101]. When the intestinal barrier is compromised, it allows for the translocation of gut bacteria, microbial products, and other antigens from the gut lumen into the circulatory system [101]. The translocation of bacteria across barriers triggers the activation of antigen-presenting cells (APCs), leading to the generation of inflammatory signaling molecules (such as TNF-alpha, IL-1, and IL-6). These molecules contribute to vascular impairment and the development of an excessive inflammatory status [13]. Moreover, HF leads to dysbiosis and results in negative changes in the composition of GM, characterized by a reduction in diversity and an increase in potentially harmful microorganisms, such as *Campylobacter* spp., *Shigella* spp., *Salmonella* spp., *Candida* spp., *Yersinia enterocolitica*, and *Chlamydia pneumoniae* [107].

Pang et al. directed a research study, using a two-sample Mendelian randomization method, to explore the potential causal link between GM and HF [108]. They gathered single nucleotide polymorphism (SNP) data related to HF and GM from the openly accessible genome-wide association analysis summary database. Their analysis uncovered seven distinct bacterial groups, which were associated with an increased risk of HF [108].

In a study by Dai et al., the authors observed that specific bacterial species were associated with the development of specific diseases: Odoribacter and Oxalobacter were most commonly associated with CVDs, even though Odoribacter was associated with a protective effect on atrial fibrillation, highlighting the difficulty in clearly understanding the exact pathways through which GM modulates cardiovascular health and the possible therapeutic implications [109]. When the intestinal barrier is not intact, bacterial components of the GM can also enter the host’s bloodstream, with a direct impact on the host’s health. Available research also shows that exosomes, for instance, play a role in the onset and progression of CVDs, such as acute coronary syndrome, atherosclerosis, HF, and myocardial ischemia–reperfusion injury [110]. Johnstone et al. discovered mesenchymal stem cell exosomes for the first time in sheep reticulocytes in 1983. They observed alterations in transferrin receptors throughout reticulocyte maturation and hypothesized that the decline in transferrin receptors within mature erythrocytes resulted from exosome secretion [111]. Specific miRNAs derived from GM carried by exosomes have been implicated in various CVDs [84]. For example, in congestive HF, an elevation in circulating levels of miR-23a, miR-23b, miR-24, miR-195, and miR-214 was observed, while levels of circulating miR-423-5p, miR-320, and miR-22 decreased. However, the exact source of these exosomal miRNAs remains unclear, posing challenges in regard to their clinical application [112]. Exosomes hold promise for the diagnosis, prognosis, and treatment of various CVDs. Due to their biological capabilities, exosomes can serve as biomarkers for diagnosing and prognosticating CVDs, thereby enhancing therapeutic strategies for these conditions [113]. Lipopolysaccharide (LPS) is a constituent of the membrane of Gram-negative bacteria residing in the intestinal tract. It can migrate into the bloodstream, leading to a condition characterized by mild endotoxemia, without systemic infection. Experimental research has shown that LPS is found in arteries affected by atherosclerosis, but not in healthy arteries. Within atherosclerotic plaques, LPS fosters a pro-inflammatory environment capable of inducing plaque instability and the formation of blood clots. This is achieved through its interaction with Toll-like receptor 4 (TLR4) across different cell types, including endothelial cells, neutrophils, monocytes, and platelets [114].

GM is also related to the development of cerebrovascular events. Animal experiments have revealed that manipulating the GM can influence stroke outcomes, as seen in germ-free mice displaying improved neurological function and decreased infarct volumes [2]. Observations indicate that stroke patients exhibit lower bacterial diversity and abundance compared to healthy individuals. For example, there is a notable decrease in the abundance of Prevotella, a genus of Gram-negative bacteria that produce short-chain fatty acids, among stroke patients. Otherwise, stroke patients have high levels of Bacteroides [3]. Zhang et al. showed that stroke patients presented lower levels of Bifidobacterium and Lactobacillus, and elevated levels of Enterobacteriaceae [4]. In a recent meta-analysis, differences in alpha and beta diversity were documented between stroke patients and the control group [5]. Other studies have determined a correlation between SCFA levels and stroke severity. Butyric acid levels have a protective effect against ischemic stroke [6]. Notably, Clostridium Butirricum has been used to modulate the GM in mouse models, with an improvement in cognitive function and a reduction in neuronal damage [7].

### 5.1. Cardiovascular Diseases and Gut Microbiota-Derived Metabolites

The metabolites deriving from GM can modulate the cardiovascular system, in a positive or negative way. GM presents two main metabolites: TMAO, produced by microbial metabolism of dietary nutrients, such as choline and carnitine, and SCFAs, produced through the fermentation of dietary fiber [115].

Blood levels of SCFAs, particularly butyrate, were correlated with a positive prognosis in patients diagnosed with HF. The SCFA levels suggested that they enhanced prognosis and decreased inflammation, observed twelve months after the initial HF episode, which were associated with the restoration of GM composition [115]. Moreover, Bartolomaeus et al., observed in murine models that SCFAs, specifically propionate, significantly mitigated fibrosis, cardiac hypertrophy, vascular dysfunction, and hypertension [116].

SCFAs are also mediators of host inflammation and immunity: they can reduce neutrophil recruitment, elevate levels of TGF-β and IL-10, and decrease levels of IL-6, IL-1β, and TNF-α, thus inhibiting the inflammatory response [117].

The GM also plays a pivotal role in the synthesis of TMAO, which is the oxidative metabolite of trimethylamine (TMA). Most of the TMA generated by gut bacteria via choline and L-carnitine metabolism enters the bloodstream and is quickly converted into TMAO by liver enzymes that contain flavin monooxygenase [117]. In a prospective observational study conducted by Zhang et al., which involved 155 patients with HF, 100 with stable diseases, and 33 healthy controls, it was found that the plasma levels of TMAO were notably higher in patients with HF compared to healthy controls [118].

While SCFAs and TMAO are the main metabolites produced by the GM, there are also others. Czibik et al., for instance, showed a pathogenic mechanism wherein elevated phenylalanine levels contribute to cardiac aging, underscoring the modulation of phenylalanine as a potential therapeutic approach for age-related cardiac impairment [119].

Several studies have indicated a significant decrease in ricinoleic acid, a metabolite of GM and the primary constituent of castor oil, among patients with chronic HF [115]. Interestingly, ricinoleic acid levels exhibited a negative correlation with bacterial communities enriched in the gut of chronic HF patients and a positive correlation with those found in the microbiota of healthy individuals without cardiovascular issues [120].

Alterations in microbiota composition can influence the bile acid pool, thereby contributing to cardiometabolic diseases. Bile acids interact with various receptors, such as the Takeda G protein–coupled receptor 5 (TGR5), the muscarinic M2 receptor, and the farnesoid X receptor (FXR) expressed on cardiomyocytes, exerting inotropic, lusitropic, and chronotropic effects [8]. The bile acid receptors FXR and TGR5 are significant in the context of heart failure. For instance, in rats, the activation of FXR receptors by secondary bile acids can improve the bile acid ratio and suppress nuclear factor kappa-light-chain-enhancer of activated B cells (NF-κB) activation, thereby averting inflammation and hypertrophy in the myocardium. Prolonged NF-κB activation results in the increased expression of the atrial natriuretic factor, contributing to cardiomyocyte enlargement. NF-κB serves as a crucial transcription factor, up-regulating numerous genes involved in inflammation, proliferation, cell differentiation, and cell death [9].

Phenylacetylglutamine is a metabolite produced by the GM from its precursor metabolite, phenylalanine [10]. Phenylacetylglutamine has been demonstrated to influence the potential for thrombosis, by interacting with G protein–coupled receptors (GPCRs) and adrenergic receptors (ADRs) [10]. This interaction enhances platelet function, resulting in hyperresponsive platelets, which can predispose individuals to myocardial infarction in coronary heart disease. Additionally, the interaction of phenylacetylglutamine with GPCRs and ADRs promotes the activation of the sympathetic nervous system, thus exacerbating HF [9]. Dietary breakdown products, including aromatic amino acids, such as tyrosine, phenylalanine, and tryptophan, as well as compounds like choline, betaine, and L-carnitine, are metabolized by gut bacteria into the precursors of recognized uremic cardiotoxins/nephrotoxins, such as p-cresol, indole, and trimethylamine, along with uremic toxins like indole acetic acid. However, as kidney function declines, levels of these metabolites increase in the bloodstream, exerting pathophysiological effects on the blood vessels, heart, and kidneys [11].

Studies conducted in human umbilical vein endothelial cells (HUVECs) and rat aortic rings have demonstrated that indoxyl sulfate (IxS) induces the expression of Nox4 mRNA and the production of reactive oxygen species (ROSs), while also reducing nitric oxide (NO) production and cell viability. These effects may collectively contribute to endothelial dysfunction [12]. The generation of reactive oxygen species (ROSs) induced by IxS has been linked to endothelial cell senescence, as demonstrated by the elevated activity of senescence-associated beta-galactosidase (SA-β-gal) [13].

P-cresyl sulfate (pCS) promotes the shedding of endothelial microparticles, which appear to be capable of selectively impairing the endothelial NO signal transduction pathway, which ends in endothelial dysfunction. Additionally, pCS may also contribute to endothelial dysfunction through the disruption of the endothelial barrier function, which is mediated by Src-induced phosphorylation of VE-cadherin [11]. Moreover, pCS may also be involved in arterial calcification. When human aortic smooth muscle cells are incubated with pCS, it induces the expression of osteoblast-specific proteins, such as alkaline phosphatase (ALP), osteopontin (OPN), core-binding factor alpha 1 (Cbfa1), and enhances ALP activity [14]. Additionally, pCS induces insulin resistance, a frequent occurrence in chronic kidney disease (CKD), by disrupting signaling pathways in skeletal muscle through the activation of ERK1/2 [11]. In Table 1, a summary of the effects of the different microorganisms and GM metabolites and their effect on CV health is provided.

### 5.2. Cardiovascular Diseases and Sex Hormones

The interactions between sex and GM, and between GM and cardiovascular health, are both well-known, thus it is reasonable to hypothesize that these three elements interact with one another. Some differences in GM composition between men and women are likely to contribute to the differences in terms of cardiovascular health [123] (Table 2).

As discussed above, men and women present different patterns of CVDs. One particularly interesting aspect is that men tend to develop CVDs earlier, while women often experience more comorbidities at an earlier stage of life, and some authors suggest this may be due to the difference in microbiota composition [124]. In a 2024 study by Garcìa-Fernandez et al., it was observed that men and women who develop congestive heart failure present different patterns of dysbiosis, but the mechanisms underlying this difference are unclear [125]. One mechanism through which the microbiota may exert a sex-specific role on cardiovascular health is through hormonal modulation [126,127].

In a study conducted on mice by Cross et al., it was observed that hormonal changes after surgical menopause increase the risk of metabolic syndrome, and that being fed a low-fat diet significantly modifies the composition of GM in mice [128]. One very interesting aspect is that the changes in hormonal levels are associated with increased gut permeability, which has been related to metabolic syndrome [129]. Increased inflammation has been observed during the menopause, during which gut permeability increases significantly [130]. Systemic inflammation is also involved in the development of type 2 diabetes [131], which is linked to early menopause, and is associated with an increased risk of developing CVDs [132]. Inflammation is also at play in the interaction between GM and visceral adipose tissue [133]: interestingly, even though women appear to have more risk factors for increased visceral adipose tissue presence, the contrary appears to be true [134], highlighting the continued lack of information.

The onset of CVDs is earlier in men, even though women more commonly develop comorbidities: one aspect that needs to be taken into consideration is the interaction between androgens and immune pathways. Testosterone can reduce systemic inflammation [122,135]; also, androgens appear to play a protective role against the development of inflammatory bowel diseases [136]. Conversely, androgen deprivation therapy appears to promote dysbiosis, further proving the importance of androgens in positive GM modulation [121]. These observations may create the impression that CVDs should affect women earlier than men, but more elements are at play than one may initially think. For instance, while it is true that reduced levels of androgens promote the proliferation of visceral adipose tissue, the same has been observed in the case of particularly high levels of these hormones [137,138]. Another interesting aspect is that visceral adipose tissue can negatively impact androgen levels [139], which perpetuates this vicious circle.

The Western diet also appears to play a role in this complex crosstalk: in a study on mice, it was observed that a high-fat diet can directly impair testosterone production through the Klk1bs/Eid3 pathway [140]. Interestingly, experiments conducted on female macaques showed that, while a Western-style diet does promote adipocyte hypertrophy, it does not promote a proinflammatory status if testosterone levels are elevated, suggesting a modulating effect of this hormone [141].

The importance of a high-fat diet on GM is also well-known: in a study by Cai et al., using a murine model, it was observed that the microbiota of mice exposed to a high-fat diet was richer in Ileibacterium and Desulfovibrio. The former has been associated with obesity, which, as discussed, can impact androgen levels [142].

However, the interaction between GM and androgens also involves direct modulation: GM can alter the androgen levels of its host, which can have negative effects on cardiovascular health [143].

The complexity of the relationship between gender, microbiota, and cardiovascular health is highlighted in the case of transgender persons: CVDs are more common in those who identify as transgender and the guidelines are not yet clear whether they should receive specific treatment in this regard [144]. The higher prevalence in this group of CVDs is likely the result of several factors, including stress and discrimination, but the role of gender-affirming therapy also needs to be considered [145]. In a study by Valentine et al., it appears that transgender youths are at a higher risk of hyperlipidemia and liver diseases when taking testosterone and a gonadotropin-releasing hormone agonist; testosterone alone is associated with obesity and hypertension, while estradiol and gonadotropin-releasing hormone agonists are not associated with the development of CVDs [146]. Yet, as noted by the authors, this population is at a higher risk of depression and the use of psychiatric drugs, which can directly impact cardiovascular health [147], and when correcting for these factors, it was observed that transgender youths undergoing gender-affirming care are only at risk of being overweight or obese. While these risks need to be taken into consideration before initiating gender-affirming care, it is also worth noting that these therapies have an overall positive impact, thus further studies are necessary to understand how to reduce health risks in this population, to further enhance the benefits of the available treatments [148].

**Table 2 biomedicines-12-01645-t002:** Differences in GM composition between men and women and different contribution in cardiovascular health.

Disease	Sex	References
Heart failure	Men > women ↑ early menopause	[127,149]
Myocardial infarction	Hypertension increases risk in women > men ↑ reparative function in women	[150,151,152]
Atrial fibrillation	↑ testosterone levels	[153]
Metabolic syndrome	↑ androgens ↓ ovarian hormones ↑ female sexual hormone-binding globulin plasma levels	[154]
Valvular disease	↑ mistreatment in women	[155]
Hypertension	Men > women ↑ vasoconstrictor response from testosterone	[44,156]

↑ high; ↓ low.

## 6. Therapeutic Implications and Future Directions

Microbiota modulation has been the subject of many studies in the context of different diseases [157,158]. The possible approaches can vary, ranging from dietary modifications, probiotics, prebiotics, and FMT, and have proven effective, particularly in regard to gastrointestinal disorders [159]. GM has been studied also in the context of CVDs, particularly given the strong interaction between dietary habits and cardiovascular health [160]. It is, indeed, well-known that high-fat diets can negatively modulate GM and promote CVDs [161], while the Mediterranean diet has a positive impact on both GM composition and cardiovascular health [162]. Yet, the complexity of the GM and its continuous evolution represent a challenge in targeting it for specific disorders at different stages of life [163]. Yet, some approaches have shown promise: for instance, polyphenols can act as probiotics, promoting healthy GM and cardiovascular benefits [149]. Another interesting approach is represented by FMT: persons with metabolic syndrome who underwent the procedure experienced improved insulin sensitivity and dyslipidemia [164], even though some authors do not agree [165]. Yet, some trials are ongoing to evaluate whether FMT could have applications in other diseases, in addition to recurring Clostridium difficile infection [166]. Other approaches to GM modulation are also being evaluated [167], but it is worth remembering that the precise mechanisms through which they work are not yet clear, thus it is necessary to proceed with caution [168].

While sex differences play a role in GM modulation, there are still no available studies on the differences in the effectiveness of specific modulating agents in men and women.

In conclusion, both sex and GM play key roles in the development, progression, and treatment of cardiovascular disease. Their interplay further contributes to influencing cardiovascular health and, while some aspects have been studied in detail, studies on the matter are still not conclusive. Understanding the interplay between sex and GM is essential to improve care, particularly for women, who are more likely to have worse experiences when interacting with healthcare services. Targeting the GM holds promise as a novel therapeutic strategy for improving cardiovascular health, with potential implications for reducing the sex disparities in CVD outcomes. Yet, available data are still scarce, and data on sex-specific approaches are mostly only available in animal models. Further research is needed to elucidate the mechanisms underlying sex-specific interactions between GM and CVDs, leading the way for personalized and precision medicine approaches in cardiovascular care.

## Figures and Tables

**Figure 1 biomedicines-12-01645-f001:**
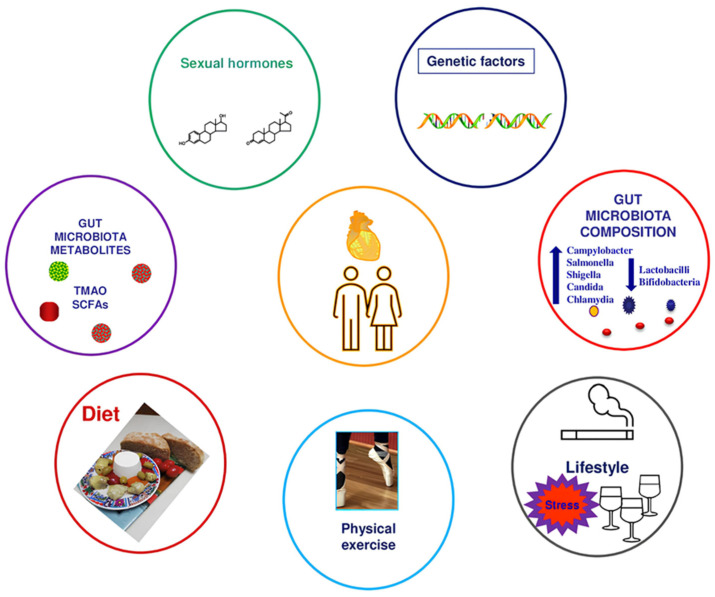
The impact of sex-related factors on cardiovascular health and disease.

**Table 1 biomedicines-12-01645-t001:** GM and CV health.

Gut Bacteria and Their Compounds	Associated Cardiovascular Diseases	References
Christensenellaceae, *Methanobrevibacter*, Ruminococcaceae	Higher lipidaemia and increased CVD risk	[84]
*Campylobacter* spp., *Salmonella* spp., *Shigella* spp., *Yersinia enterocolitica*, *Candida* spp., *Chlamydia pneumoniae*	Heart failure	[119]
Lipopolysaccharide	Atherosclerosis	[114]
Low levels of *Prevotella*, *Bifidobacterium*, and *Lactobacillus*	Stroke	[1]
TMAO	Heart failure	[121]
Low levels of ricinoleic acid	Heart failure	[122]
Phenylacetylglutamine	Potential thrombosis exacerbating heart failure	[2,3]
P-cresol, Indole, and Trimethylamine	Pathophysiological effects on the blood vessels, heart, and kidneys	[4]

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
