# Peer review of "Sex Differences in Cardiovascular Diseases: Exploring the Role of Microbiota and Immunity"

_biomedicines, 2024, doi:10.3390/biomedicines12081645_

Round 1

Reviewer 1 Report

Comments and Suggestions for Authors

The article covers a significant topic but lacks consistency. Appropriate and key studies are included, relevant and recent, and the cited sources are referenced correctly. The paper is comprehensive, but the logical flow should be improved, and the data should be presented critically.

However, there are some specific comments on the weaknesses of the article and what could be improved:

Major points

1. One of the major drawbacks is not including the genetic factors associated with gender and CVDs

2. The second section resembles the title, but it should be precise

3. Section 3 starts with microbiota and then with factors not related to the microbiota. Probably, the structure of section 2 and 3 should be revised.

4. Sections 4, 5 and 6 bring even more confusion. I suggest for the authors to consider 2-3 subtitles with sub-subtitles for each

Minor points

1. Subsections 3.2 and 3.3. are entitled 3.2. Figures, Tables and Schemes

2. Figure 1 should include genetic factors

3. References have doubled numbering

Author Response

Dear Editor of Biomedicines

First, my coauthors and I would like to thank You sincerely for this opportunity of cooperation. We profoundly thank the reviewers for the comments and useful suggestions aimed at improving the paper. We thank You for your constructive critique and we hope the review process has led to an improved manuscript. If additional changes are warranted, we will make them. 

We hope that this revised version of our manuscript may now be found suitable for publication. 

This is a point-by-point list of changes made in the paper:

Reviewer 1

The article covers a significant topic but lacks consistency. Appropriate and key studies are included, relevant and recent, and the cited sources are referenced correctly. The paper is comprehensive, but the logical flow should be improved, and the data should be presented critically.

However, there are some specific comments on the weaknesses of the article and what could be improved:

Major points

  1. One of the major drawbacks is not including the genetic factors associated with gender and CVDs
    We have modified the paragraph 2 including the genetic factors associated with gender and CVDs as requested.
  2. The second section resembles the title, but it should be precise
    We have modified the second section, as suggested.
  3. Section 3 starts with microbiota and then with factors not related to the microbiota. Probably, the structure of section 2 and 3 should be revised.
    We have revised the structure of these sections, as suggested.
  4. Sections 4, 5 and 6 bring even more confusion. I suggest for the authors to consider 2-3 subtitles with sub-subtitles for each
    We have modified them, as suggested.

Minor points

  1. Subsections 3.2 and 3.3. are entitled 3.2. Figures, Tables and Schemes
    We have corrected the mistakes, as suggested
  2. Figure 1 should include genetic factors
    We have modified the figure, as suggested
  3. References have doubled numbering
    We have revised all the references, as suggested

We thank You for your constructive critique and we hope the review process has led to an improved manuscript.

If additional changes are warranted, we will make them.

We hope that this revised version of our manuscript may now be found suitable for publication.

Sincerely,

Mario Caldarelli

Reviewer 2 Report

Comments and Suggestions for Authors

Franza et al. propose a narrative review summarising evidence deriving from the analysis of sex differences in cardiovascular disease, more specifically focusing on the effects of microbiota and immunity. The selected topic is quite current, and I find this to be a timely piece.

Other than some minor inaccuracies (e.g., "synthesised" instead of "summarising" row 70 page 2), the manuscript is well written and well organised.

The subtitle 3.2 and 3.3 need to be corrected (they read not as "Figures, Tables and Schemes"). Consider checking the capital letters for table 1 in the fields corresponding to "Phenylacetylglutamine", and "P-cresol, indole, and trimethyla-mine". Also, row 501 it should read "gender-affirming" rather than "sex-affirming". 

Author Response

Dear Editor of Biomedicines

First, my coauthors and I would like to thank You sincerely for this opportunity of cooperation. We profoundly thank the reviewers for the comments and useful suggestions aimed at improving the paper. We thank You for your constructive critique and we hope the review process has led to an improved manuscript. If additional changes are warranted, we will make them. 

We hope that this revised version of our manuscript may now be found suitable for publication. 

This is a point-by-point list of changes made in the paper:

Reviewer 2

The article covers a significant topic but lacks consistency. Appropriate and key studies are included, relevant and recent, and the cited sources are referenced correctly. The paper is comprehensive, but the logical flow should be improved, and the data should be presented critically.

However, there are some specific comments on the weaknesses of the article and what could be improved:

  • Franza et al. propose a narrative review summarising evidence deriving from the analysis of sex differences in cardiovascular disease, more specifically focusing on the effects of microbiota and immunity. The selected topic is quite current, and I find this to be a timely piece.

    Other than some minor inaccuracies (e.g., "synthesised" instead of "summarising" row 70 page 2), the manuscript is well written and well organised.

    The subtitle 3.2 and 3.3 need to be corrected (they read not as "Figures, Tables and Schemes"). Consider checking the capital letters for table 1 in the fields corresponding to "Phenylacetylglutamine", and "P-cresol, indole, and trimethyla-mine". Also, row 501 it should read "gender-affirming" rather than "sex-affirming".

Thank you for the feedback. We have made the changes as suggested.

We thank You for your constructive critique and we hope the review process has led to an improved manuscript.

If additional changes are warranted, we will make them.

We hope that this revised version of our manuscript may now be found suitable for publication.

Sincerely,

Mario Caldarelli

Round 2

Reviewer 1 Report

Comments and Suggestions for Authors

The authors improved the paper significantly according to the recommendations. 

Author Response

Thank you for the feedback.

Best regards, Mario Caldarelli